# Study on the Impact of Corporate ESG Performance on Green Innovation Performance—Evidence from Listed Companies in China A-Shares

**Jing Zhang and Ziyang Liu ***

Graduate School, Kyonggi University, Suwon 16227, Republic of Korea; zhangjing@kgu.ac.kr
* Correspondence: victor@kgu.ac.kr

**Abstract:** With the establishment of China's "dual carbon" target and the promotion of high-quality development strategy, the role of green innovation has become increasingly important. Corporate ESG innovation, as a guiding principle for companies to practice sustainable development and an important signal for evaluating their environmental and social responsibilities as well as corporate governance level, deserves in-depth research on its impact on green innovation performance. This paper empirically analyzes the green innovation effect of corporate ESG (Environmental, Social and Governance) performance using Chinese A-share listed companies as a sample from 2009 to 2021. The research shows that corporate ESG performance can enhance green innovation performance. Mechanism analysis reveals that ESG performance mainly improves green innovation performance by alleviating financing constraints and enhancing human capital. Further research shows that all three sub-dimensions of ESG performance contribute to improving green innovation performance, with the strongest effect observed in corporate governance performance. ESG performance not only enhances strategic green innovation performance and independent green innovation performance but also improves substantial green innovation performance and collaborative green innovation performance. Therefore, the government should improve the ESG information disclosure system, increase support for companies with excellent ESG performance, and improve local talent policies to attract high-quality green innovation talents. Investors should incorporate ESG performance into their decision-making and strengthen the identification and use of ESG information. Companies should formulate ESG strategies, increase relevant investments, prioritize corporate governance improvement, and enhance the quality of ESG information disclosure through various means.

**Keywords:** ESG performance; green innovation; financing constraints; human capital level





## 1. Introduction

With the establishment of China's dual carbon goals and the proposal and implementation of a high-quality development strategy, the green transformation of the economy and society and sustainable development have become the mainstream consensus. Energy conservation and emission reduction, energy transition, and resource recycling are necessary and key approaches to achieving the strategic goals mentioned above [1–3]. Multidimensional green innovation, represented by energy transition and processing technology, waste management, and transportation, among others, is an effective and efficient way to promote green transformation [4–6].

As the main entities involved in energy production, utilization, and waste disposal, enterprises need to engage in green innovation as a necessary means to achieve China's dual carbon goals. By promoting clean energy, improving energy efficiency, and optimizing energy structure, enterprises can reduce carbon emissions and contribute to carbon neutrality [7,8]. Green innovation by enterprises is also an important pathway to drive high-quality development. By improving product design, enhancing production efficiency, and adopting various measures, enterprises can simultaneously increase both economic

and environmental benefits, thereby achieving sustainable development. Furthermore, green innovation can enhance a company's competitiveness. If enterprises can launch green products that meet consumers' environmental demands, they can establish a good corporate image, gain more market share, and achieve long-term development [9,10].

However, similar to general innovation activities, enterprises need significant resources, capabilities, and willingness to engage in green innovation. Among these factors, financing constraint is one of the key challenges that hinder the improvement of green innovation performance. Compared to general innovation activities, financing constraint has a more pronounced inhibitory effect on green innovation [11–13]. Human capital stock is another crucial factor influencing whether enterprises choose to engage in green innovation. It serves as the main body for enterprise green innovation activities and the source of necessary knowledge and skills. Therefore, the ability to alleviate financing constraints and attract, retain, and motivate talents for green innovation has become a necessary path for enterprises to engage in green innovation. Due to the existence of information asymmetry, enterprises should actively send positive signals and convey favorable information in the lending market, capital market, and labor market. As an important signal for evaluating corporate environmental and social responsibility and corporate governance level, ESG performance should play a greater role in the process of enterprise resource acquisition. With the accelerated construction of ESG information disclosure systems by regulatory authorities and the introduction of policies and regulations rooted in the new development concept, which are in line with the current stage of development for Chinese enterprises, it is foreseeable that the ESG information disclosure system will be continuously improved [14].Therefore, it is of great value to explore the green innovation effects of corporate ESG performance and investigate the underlying mechanisms.

This study empirically analyzes the green innovation effects of ESG performance of Chinese A-share listed companies from 2009 to 2011 and examines their influencing mechanisms. The main marginal contributions of this study are as follows: Firstly, despite some achievements in research on the economic consequences of corporate ESG performance and the influencing factors on green innovation performance, there is a lack of relevant studies exploring the relationship between the them [15–17]. Therefore, this study contributes to the literature on the economic consequences of ESG performance and the factors influencing green innovation performance, particularly in the realm of the relationship between ESG performance and green innovation performance, thus promoting further research development. Secondly, green innovation performance is an area of continuous focus for enterprises, given the requirements of dual carbon goals, the new development concept, and responsible investment. Green innovation is the necessary path for enterprises to contribute to achieving the dual carbon goals and implementing the new development concept, as well as a catalyst for enhancing the sustainability of enterprises' competitive advantages and improving environmental, social, and economic benefits. Therefore, this study can provide decision-makers and stakeholders with the foundational and background knowledge necessary for relevant decision-making, reduce irrational decision-making, and support enterprises' ESG practices and green innovation activities. Lastly, with the implementation of China's dual carbon goals strategy, the new development strategy, and the attention given by government departments to the issue of sustainable development of listed companies, promoting high-quality development among listed companies through policy design and institutional establishment has become an important question. Therefore, this study can contribute to answering some questions in the process of listed company regulation and information disclosure system construction, thereby improving the relevance and effectiveness of relevant policies and regulatory systems.

## 2. Literature Review and Research Assumptions
### 2.1. Enterprise ESG Performance and Green Innovation Performance

The Firstly, based on the content of the ESG concept system, it can be inferred that the better a company performs in terms of environmental responsibility, the more it val-

ues environmental protection. This suggests that the company is more likely to integrate the concept of sustainable development into its mission, strategy, and culture, as well as increase relevant resource allocation. Studies have shown that a company's commitment to green strategies, green organizational culture, green organizational identification, and environmental values can effectively promote green innovation [18–20]. Additionally, a better environmental performance of a company indicates a richer knowledge base and widespread presence of green knowledge within the organization, which is essential for organizational innovation [21]. Therefore, the accumulation and sharing of such green knowledge are advantageous for green innovation. Furthermore, a company's environmental performance reflects the high environmental awareness and attitude of its management, driving green innovation [22]. The Porter hypothesis suggests that strict regulations can stimulate innovation, enabling companies to achieve dual objectives of environmental protection and improved business performance. Building upon this, Wong (2013) developed a comprehensive model to explore the determinants of green innovation [23]. Tang et al. (2020) empirically analyzed the impact of environmental regulations, represented by China's "Eleventh Five-Year Plan", on the green innovation performance of businesses [24]. Additionally, Cai et al. (2020) investigated the influence of direct environmental regulations on green technological innovation of listed companies in highly polluting industries in China, from the perspective of institutional monitoring theory, using a panel-Poisson fixed effects model [25]. Xu et al. (2021) conducted a survey on 62 energy-saving and environmental protection listed companies in the Chinese market from 2013 to 2018, and verified through PSM-DID analysis that environmental performance significantly enhances innovation performance of companies. Heterogeneity analysis further revealed that this conclusion holds particularly true for highly polluting companies [26].

Furthermore, the better a company performs in terms of corporate social responsibility, the more it indicates that the company is attentive to and safeguarding the interests and needs of stakeholders such as suppliers, customers, and employees. This, in turn, promotes the green innovation performance of the company. Awawdeh et al. (2021) estimated the relationship between corporate social responsibility (CSR) and corporate environmental performance in energy companies operating in Egypt using an empirical model [27]. They further evaluated the mediating effect to assess the role of green financing in improving corporate environmental performance. Hussain (2022) examined the impact of CSR on environmental performance using data collected from ten large industrial organizations operating in Lahore, Pakistan [28]. Naveed (2022) employed the 2SLS method to study the effects of board gender diversity, CSR disclosure, and corporate green innovation performance [29]. According to stakeholder theory, this fosters trust-based relationships between the company and its stakeholders, making them more willing to share subtle information regarding utility functions. Consequently, this facilitates targeted innovation to meet the needs of stakeholders [30]. For example, when companies gain accurate and timely insights into the demand for green products from customers, targeted green technology and product development can capitalize on market opportunities, resulting in a higher market share and improved green innovation performance. Moreover, stakeholders are more willing to collaborate with companies that have a higher level of ESG performance, making it easier for companies to acquire scarce resources necessary for innovation. For instance, fulfilling social responsibilities helps improve supply chain relationships and promotes collaboration, enhancing green innovation performance [31]. This enables companies to acquire external knowledge shared by stakeholders, complementing internal knowledge and facilitating knowledge exchange and sharing, thereby promoting green innovation performance [32,33].

Lastly, a company's good corporate governance reflects a higher level of governance. According to the principal-agent theory, the agency problem hinders research and development innovation in the presence of high objective and subjective risks. For example, shorter tenure of senior executives, less equity incentives, and risk aversion among agents aggravate this problem [34]. Therefore, mitigating the agency problem and reducing agency

costs become crucial measures for enhancing green innovation in companies. Specifically, ESG practices can enhance corporate governance through three channels, thereby promoting green innovation: (1) improving board governance by, for instance, increasing board size [35], the proportion of independent directors [36,37], and enhancing gender and age diversity [38,39]; (2) enhancing management incentive arrangements, such as promoting equity incentive plans [40] and (3) improving shareholder supervision mechanisms, such as involving professional institutional investors [41].

To sum up, this study puts forward the following hypothesis:

**H1:** *Enterprise ESG performance has a positive impact on green innovation performance.*

### 2.2. The Mediating Role of Financial Constraints

On one hand, the main causes of financial constraints are the information asymmetry and agency conflicts between companies and fund holders. Based on the signaling theory, companies should actively release positive signals to alleviate information asymmetry and position themselves favorably. The ESG performance of companies serves as an important signal for banks, investors, and other stakeholders in the credit and capital markets. According to the principal-agent theory, companies should proactively seek supervision and incentive arrangements to reduce the risk premium sought by external financiers due to information asymmetry and agency problems. Research has shown that good corporate ESG performance provides financial and non-financial information, which helps fund holders better understand and supervise the company. Therefore, companies with good ESG performance can obtain lower-cost debt financing and equity financing by leveraging signaling and governance effects [42], thereby alleviating financial constraints [43].

On the other hand, green innovation is a capital-intensive activity, making financial constraints a primary "threshold" hindering companies from enhancing their own green innovation capabilities [44]. This obstruction is more severe for green innovation compared to general innovation, particularly in terms of research and development of green invention patents and energy resource-efficient patents [21]. Therefore, companies require large, stable, and long-term external financing to mitigate the financial risks associated with innovation investments. Based on this, the following hypothesis is proposed:

**H2:** *Corporate ESG performance promotes green innovation performance by alleviating financial constraints.*

### 2.3. The Mediating Role of Human Capital Level

The green knowledge, skills, and capabilities of human capital can provide companies with new knowledge and technologies for green innovation. It can also integrate green innovation into the traditional knowledge base and values, continuously promoting green innovation [45]. On one hand, compared to general innovation activities, green innovation in companies requires more external information and specialized knowledge, particularly in understanding and solving environmental issues. Therefore, the absorptive capacity inherent in human capital facilitates the identification and utilization of external innovation opportunities, contributing to the mastery and application of new knowledge in product and technological development [46]. On the other hand, the implementation of green innovation in companies relies on employees with high innovation potential, who possess environmental ethical values. Designing reasonable and effective incentive systems helps stimulate employee green innovation capabilities [47].

Firstly, according to stakeholder theory, the fulfillment of ESG responsibilities in companies meets a variety of employee needs, fosters employee identification, facilitates collective creation among employees, reduces job insecurity, and enhances employee engagement, thus playing a role in employee motivation and retention [48], ultimately promoting innovative behavior among employees. Additionally, it expands channels for employee training, education, and learning, contributing to enhancing the quality of human

resources and supporting green innovation [10]. Secondly, based on signaling theory, the ESG performance of companies also serves as important information transmission in the labor market, attracting higher-quality potential applicants with strong environmental ethical values to companies with higher ESG levels [49]. In summary, corporate ESG performance helps develop human capital and supports green innovation. Based on this, the following hypothesis is proposed:

**H3:** *Corporate ESG performance promotes green innovation performance by enhancing the level of human capital.*

### 3. Research Design Sample Selection and Data Source

*3.1. Data Source*

This study uses Chinese A-share listed companies from 2009 to 2021 as the initial research sample and conducts the following processing based on research needs: (1) Considering the industry specificity, we exclude samples from the financial and insurance sectors. (2) In order to ensure data authenticity and accuracy, we exclude samples from companies classified as ST, ST*, or PT. (3) To guarantee data comparability and effectiveness, we exclude observations with missing values for the main explanatory and dependent variables. (4) In order to address the impact of outliers, we applied trimming to all variables. To minimize estimation error caused by sample loss, we cautiously chose to perform trimming at a significance level of 1%. During the data processing, we examined scatter plots of text variables and identified a few extreme outliers. It is worth mentioning that these outliers did not have any impact on the main conclusions of our study [50]. Although there were slight differences in the regression coefficients between the trimmed and untrimmed results, there were no differences in terms of sign and significance of the coefficients.. Finally, the study obtains an unbalanced panel dataset with 3877 listed companies, comprising a total of 30,970 firm-year observations. Considering the coverage and time span of ESG ratings, this study selects the CICC(China International Capital Corporation Limited) ESG ratings as a proxy for corporate ESG performance. Green patent data is obtained from the China Research Data Services Platform, while financial data and other company characteristic data are sourced from the CSMAR database (China Stock Market & Accounting Research Database).

*3.2. Variable Definitions*

3.2.1. Dependent Variable

The dependent variable in this study is the green innovation performance of enterprises, represented by the symbol GIP (Green Innovation Performance). Existing research mostly uses the level of green innovation's knowledge output or expected output as proxy variables [13,51]. Considering the availability and accuracy of the data, this study selects the level of knowledge output of green innovation, specifically the number of green patent applications, as the proxy variable for the green innovation performance of enterprises.

3.2.2. Independent Variable

The independent variable in this study is the ESG performance of enterprises, represented by the symbol ESG. Existing research mostly uses ESG ratings and scores published by third-party ESG rating agencies, such as MSCI (Morgan Stanley Capital International Index, Manhattan, NY, USA) ESG and Thomson Reuters ESG internationally, and Huazheng ESG domestically. Although the foreign rating systems are more comprehensive, considering that the sample in this study is Chinese A-shares, the Huazheng ESG rating system, which is more in line with the Chinese reality and has a longer coverage period, is selected as the proxy variable for the ESG performance of enterprises.

### 3.2.3. Mediating Variables

The mediating variables in this study are financing constraints and human capital level, represented by the symbols SA and HP, respectively. Firstly, existing research mostly measures the degree of financing constraints of enterprises using methods such as internal cash flow model, composite index, and management's subjective perception. Considering the availability of data, the objectivity of measurement, and the exogeneity of variables, this study selects the SA index as the proxy variable for the financing constraints of enterprises. Secondly, existing research mainly uses employee education, knowledge, and skills to measure the level of human capital. Considering the observability of variables and the availability of data, this study selects the number of employees with a bachelor's degree or above as the proxy variable for the human capital level of enterprises.

### 3.3. Model Specification

Refer to existing literature [17], in order to examine hypothesis H1, this study constructs the following basic model:

$$GIP_{i,t+1} = \beta_0 + \beta_1 ESG_{i,t} + \beta_2 Controls_{i,t} + Industry + Year + \varepsilon_{i,t}$$

In the model, the subscripts *i* and *t* represent individual samples and years, respectively. The dependent variable GIP represents the green innovation performance of enterprises, which is proxied by the natural logarithm of the sum of green invention patents and utility model patents independently applied for by listed companies, plus 1. A one-period lag is included to control for lagged effects and reverse causality that may lead to endogeneity.

The independent variable ESG represents the assigned ESG ratings of listed companies (ranging from CCC to AAA, assigned values from 1 to 9, respectively).

Referring to existing research, this article selects corporate characteristics such as financial condition, corporate governance, and corporate nature as control variables [52,53]. Additionally, industry fixed effects (Industry) and annual fixed effects (Year) are included. The error term is represented by $\varepsilon$, and further details are provided in Table 1.

**Table 1.** Variable Declaration.

| Variable Categories | The Variable Name | Variable Symbol | Variable Declaration |
| --- | --- | --- | --- |
| Be explained variable | Green innovation performance | lnGIP | Add 1 green patent applications take logarithm |
| Explanatory variables | ESG performance | ESGRating | China the ESG rating assignment 1–9 |
| Intervening variable | Financing constraints | SA | The absolute value of SA index |
| | The human capital level | HP | Bachelor degree or above the number of employees of logarithm |
| Control variables | Fixed assets ratio | Fix | Tangible assets/total assets |
| | Asset-liability ratio | Lev | The total debt/total assets |
| | Cash levels | Cash | The sum of cash and its equivalents/total assets |
| | The market value | Tobin | The company's market value/asset replacement cost |
| | profitability | Roa | Net income/total assets |
| | The enterprise growth | Growth | Main business revenue growth/main revenue last year |
| | Ownership concentration | Top1 | The listed company's largest shareholder shareholding |
| | Management ownership | Msahre | Management ownership/total equity |
| | The board of directors meeting times | Meeting | The board of directors of the listed company |
| | The independent directors proportion | Indp | The total number of dong people alone/the board of directors |
| | Directors and managers of situation | Duality | Existence of Duality = 1, and vice in 0 |
| | The enterprise scale | lnsize | At the end of the total assets of the natural logarithm |

**Table 1.** *Cont.*

| Variable Categories | The Variable Name | Variable Symbol | Variable Declaration |
|---|---|---|---|
| | Enterprise age | lnage | Set up the fixed number of years the natural logarithm |
| | Property rights | Govn | State-owned enterprises (soes) = 1, and vice in 0 |
| | year | Year | Annual fixed effect |
| | industry | Indsutry | Industry fixed effects |

## 4. Empirical Analysis

### 4.1. Descriptive Statistics

First, a descriptive statistical analysis is conducted on all variables. The results are shown in the Table 2. The analysis reveals that the level of green innovation in Chinese A-share listed companies is generally low, with significant variations. The overall ESG performance is average, but with substantial differences among companies. Companies, in general, face financing constraints, and there is room for improvement in the human capital level of some enterprises.

**Table 2.** Descriptive Statistics of Variables.

| Variable | Observations | Mean | Standard Deviation | Minimum | Maximum |
|---|---|---|---|---|---|
| lnGIP | 30,970 | 0.878 | 1.157 | 0 | 7.319 |
| ESG | 30,970 | 4.107 | 1.1 | 1 | 8 |
| SA | 30,953 | 3.76 | 0.275 | 0.271 | 5.646 |
| HP | 19,242 | 6.355 | 1.387 | 1.099 | 12.507 |
| Fix | 30,962 | 0.216 | 0.166 | 0 | 0.971 |
| Lev | 30,962 | 0.458 | 1.056 | −0.195 | 138.378 |
| Cash | 30,941 | 0.199 | 0.148 | 0 | 1 |
| Tobin | 30,260 | 2.803 | 86.016 | 0.641 | 14,810.31 |
| Roa | 30,962 | 0.039 | 0.737 | −51.947 | 108.366 |
| Growth | 30,911 | 5.353 | 770.407 | −1 | 135,000 |
| Top1 | 30,953 | 34.687 | 15.173 | 0.29 | 89.99 |
| Mshare | 29,934 | 0.133 | 0.203 | 0 | 4.219 |
| Meeting | 29,977 | 9.766 | 4.111 | 0 | 58 |
| Indp | 30,950 | 0.375 | 0.056 | 0 | 1 |
| Duality | 30,452 | 0.272 | 0.445 | 0 | 1 |
| lnsize | 30,941 | 22.152 | 1.386 | 11.348 | 28.636 |
| lnage | 30,937 | 2.063 | 0.903 | 0 | 3.466 |
| Govcon | 29,192 | 0.398 | 0.49 | 0 | 1 |

### 4.2. Correlation Analysis

Performing correlation analysis before conducting the baseline regression, the results are shown in the table below. The analysis indicates that the multicollinearity issue among variables is acceptable, and the correlation among the primary explanatory variables meets the requirements. The selection of variables is relatively appropriate. Partial Results of Table 3 Correlation Analysis.

### 4.3. Regression Result Analysis

Table 4 presents the baseline regression results for the impact of ESG performance on corporate green innovation performance. The coefficient of ESG (column 1) is shown to be 0.0319, significant at the 1% level. This indicates that an improvement in ESG rating of the company leads to an average increase of approximately 3.19% in the number of green invention patents and utility model patents in the following year, providing preliminary empirical evidence for the theoretical hypothesis. Controlling for other variables, the age of the company (lnage) has a negative impact on green innovation performance. This may be

due to the fact that as a company ages and reaches a mature stage, its operations become more stable and follow a certain path dependence, leading to a lack of innovation willingness. Profitability (Roa) and company size (lnsize) have a positive impact on corporate green innovation performance. One possible explanation for this is that companies with stronger profitability and larger size have more innovation resources, resulting in higher levels of innovation input and output. This validates our hypothesis 1, which aligns with the conclusions of previous studies.

**Table 3.** Correlation analysis results (part).

| Variables | (1) | (2) | (3) | (4) |
|---|---|---|---|---|
| (1) lnGIP | 1 | | | |
| (2) ESG | 0.153 *** | 1 | | |
| (3) SA | −0.013 ** | −0.052 *** | 1 | |
| (4) HP | 0.467 *** | 0.223 *** | −0.095 *** | 1 |
| (5) Fix | −0.065 *** | −0.065 *** | −0.013 ** | 0.049 *** |
| (6) Lev | 0.015 *** | −0.060 *** | −0.084 *** | 0.227 *** |
| (7) Cash | −0.056 *** | 0.108 *** | −0.150 *** | −0.090 *** |
| (8) Tobin | −0.009 | −0.018 *** | −0.063 *** | −0.201 *** |
| (9) Roa | −0.002 | 0.016 *** | −0.007 | −0.005 |
| (10) Growth | −0.005 | −0.011 * | 0.003 | 0.003 |
| (11) Top1 | 0.010 * | 0.102 *** | −0.147 *** | 0.162 *** |
| (12) Mshare | −0.039 *** | 0.084 *** | −0.175 *** | −0.209 *** |
| (13) Meeting | 0.132 *** | 0.015 *** | 0.061 *** | 0.185 *** |
| (14) Indp | 0.045 *** | 0.072 *** | −0.047 *** | 0.044 *** |
| (15) Duality | 0 | −0.009 * | −0.074 *** | −0.112 *** |
| (16) lnsize | 0.409 *** | 0.230 *** | 0.076 *** | 0.570 *** |
| (17) lnage | 0.074 *** | −0.122 *** | 0.411 *** | 0.245 *** |
| (18) Govcon | 0.060 *** | 0.054 *** | 0.089 *** | 0.318 *** |

\* $p < 0.1$, \*\* $p < 0.05$, \*\*\* $p < 0.01$.

**Table 4.** Baseline Regression Results.

| | (1) |
|---|---|
| | **F.lnGIP** |
| ESG | 0.0319 *** |
| | (4.30) |
| Fix | −0.0775 |
| | (−0.86) |
| Lev | 0.0203 |
| | (1.03) |
| Cash | −0.0659 |
| | (−1.00) |
| Tobin | 0.00677 ** |
| | (3.12) |
| Roa | 0.0695 *** |
| | (4.30) |
| Growth | 0.000000181 |
| | (0.37) |
| Top1 | −0.00169 |
| | (−1.35) |
| Mshare | −0.100 |
| | (−1.14) |
| Meeting | 0.00369 |
| | (1.89) |
| Indp | −0.0132 |
| | (−0.08) |



**Table 4.** *Cont.*

|  | (1) |
|---|---|
|  | **F.lnGIP** |
| Duality | 0.0269 |
|  | (1.17) |
| lnsize | 0.265 *** |
|  | (13.03) |
| lnage | −0.0842 ** |
|  | (−3.23) |
| Govcon | 0.0655 |
|  | (1.31) |
| _cons | −5.414 *** |
|  | (−10.66) |
| Industry | Yes |
| Year | Yes |
| N | 24,262 |
| R2 | 0.207 |

*t statistics in parentheses. ** $p < 0.05$, *** $p < 0.01$.*

### 4.4. Robustness Test

#### 4.4.1. Endogeneity Handling

The baseline regression has addressed endogeneity issues caused by reverse causality by lagging the dependent variable. To further address other potential endogeneity concerns, this study attempts to construct instrumental variables. Following previous research, the average ESG rating of firms in the same year, industry, and region (Mesg) is used as the instrumental variable for corporate ESG performance [52]. Table 5 presents the results of the two-stage instrumental variable regression.

**Table 5.** Results of Two-Stage Instrumental Variable Regression.

|  | (1) | (2) |
|---|---|---|
|  | **ESG** | **F.lnGIP** |
| ESG |  | 0.0696 ** |
|  |  | (3.20) |
| Mesg | 0.516 *** |  |
|  | (33.07) |  |
| _cons | −3.539 *** |  |
|  | (−7.81) |  |
| Controls | Yes | Yes |
| Industry | Yes | Yes |
| Year | Yes | Yes |
| N | 27,095 | 23,900 |
| R2 | 0.136 | 0.206 |
| Cragg-Donald Wald F | 1924.698 |  |

*t statistics in parentheses. ** $p < 0.05$, *** $p < 0.01$.*

Column (1) shows the results of the first-stage regression, where the coefficient of Mesg is significant and positive. The C-DWF statistic indicates no weak instrument problem, and the number of instruments matches the number of endogenous variables, indicating no overidentification. Column (2) displays the results of the second-stage regression, which align with the baseline results, thus providing further validation for research hypothesis H1 through endogeneity handling.

#### 4.4.2. Replacement of Explanatory Variables

As the China Securities ESG rating system provides scores to listed companies, which can be divided into different grades, ESG scores contain more information content than

ratings. Therefore, this study replaces the ESG rating with the natural logarithm of the China Securities ESG score (lnSc) to test the robustness of the baseline regression results. The results are shown in column (1). From the results, it can be observed that the coefficient of lnSc is 0.485, significantly positive. Through the robustness test, further support for research hypothesis H1 is provided.

### 4.4.3. Replacement of Estimation Model

Due to the count nature of corporate green innovation patents and the presence of excessive zero values, as well as the high dispersion with the standard deviation much greater than the mean, this study employs the panel negative binomial fixed effects model, which is more suitable for such data, to conduct the robustness test. In this case, the dependent variable is the number of green patents applied by listed companies. The results are presented in column (2), where the coefficient of ESG on GIP is significantly positive. Through the robustness test, further support for research hypothesis H1 is provided.

### 4.4.4. Replacement of Dependent Variable

Considering the gap between green innovation patent applications and authorizations, as stricter procedures are required for authorized patents, the granting of a patent implies a certain practical application value. Therefore, this study replaces the number of green innovation patent applications with the number of authorized green innovation patents lagged by two periods as the dependent variable to test whether ESG performance has a long-term impact on green innovation performance. The results, shown in column (3), indicate that the coefficient of ESG is significantly positive. This suggests that corporate ESG performance has a long-term impact on green innovation performance and provides further support for research hypothesis H1.

### 4.4.5. Control for Exogenous Shocks

Starting from 1 January 2018, China implemented the Environmental Protection Tax Law, which effectively promotes corporate green innovation activities [54]. In order to eliminate the influence of this exogenous shock, this study selects a sample from 2009 to 2017 and conducts regression analysis again. The results are shown in column (4) of Table 6. It can be observed that the coefficient of ESG is still significantly positive, although slightly lower compared to the baseline regression results. Overall, the results continue to support hypothesis H1 through robustness testing.

**Table 6.** Robustness Test Results.

| | (1) | (2) | (3) | (4) |
|---|---|---|---|---|
| | F.lnGIP | F.GIP | F2.lnGg | F.lnGIP |
| lnSc | 0.485 *** | | | |
| | (4.41) | | | |
| ESG | | 0.0403 *** | 0.0145 * | 0.0299 *** |
| | | (4.33) | (2.10) | (3.65) |
| _cons | −7.149 *** | −6.552 *** | −2.389 *** | −4.032 *** |
| | (−11.31) | (−19.12) | (−5.44) | (−7.89) |
| Controls | Yes | Yes | Yes | Yes |
| Industry | Yes | Yes | Yes | Yes |
| Year | Yes | Yes | Yes | Yes |
| N | 24,262 | 20,337 | 21,234 | 15,934 |
| R2 | 0.207 | 0.287 | 0.189 | 0.156 |

*t statistics in parentheses. * $p < 0.1$, *** $p < 0.01$.*

### 4.5. Mechanism Analysis

4.5.1. Mediation Effect of Financial Constraints

In this study, the absolute value of the SA index is selected as a proxy variable for financial constraints [46]. The following model is constructed to test research hypothesis H2:

$$SA_{i,t+1} = \beta_0 + \beta_1 ESG_{i,t} + \beta_2 Controls_{i,t} + Industry + Year + \varepsilon_{i,t}$$
$$GIP_{i,t+1} = \beta_0 + \beta_1 ESG_{i,t} + \beta_2 SA_{i,t} + \beta_3 Controls_{i,t} + Industry + Year + \varepsilon_{i,t}$$

The regression results for the mediation effect of financial constraints are shown in columns (1) and (2) of Table 7. In column (1), the coefficient of ESG is significantly negative, indicating that enhancing ESG performance mitigates financial constraints. In column (2), the coefficient of financial constraints is significantly negative, suggesting that financial constraints impede corporate green innovation performance. Furthermore, the coefficient of ESG performance remains significantly positive, although lower in magnitude compared to the baseline regression. This indicates that the direct effect of ESG performance on green innovation performance is smaller than the total effect after incorporating SA, indicating a partial mediation effect of alleviating financial constraints. This validates our hypothesis 2, which aligns with the conclusions of previous studies [21,41].

**Table 7.** The results of the mediation effect test.

|  | (1) | (2) | (3) | (4) |
|---|---|---|---|---|
|  | F.SA | F.lnGIP | F.HP | F.lnGIP |
| ESG | −0.00346 *** | 0.0277 *** | 0.0348 *** | 0.0270 ** |
|  | (−4.72) | (3.75) | (6.97) | (2.71) |
| SA |  | −1.044 *** |  |  |
|  |  | (−7.58) |  |  |
| HP |  |  |  | 0.177 *** |
|  |  |  |  | (5.74) |
| _cons | 3.838 *** | −1.657 ** | −6.766 *** | −4.228 *** |
|  | (35.29) | (−2.75) | (−13.14) | (−5.73) |
| Controls | Yes | Yes | Yes | Yes |
| Industry | Yes | Yes | Yes | Yes |
| Year | Yes | Yes | Yes | Yes |
| N | 24,261 | 24,262 | 15,919 | 14,554 |
| R2 | 0.824 | 0.213 | 0.537 | 0.183 |

*t statistics in parentheses.* ** $p < 0.05$, *** $p < 0.01$.

4.5.2. Mediation Effect of Human Capital Level

In this study, the natural logarithm of the number of employees with undergraduate or higher education is selected as a proxy variable for human capital level in the company. The following model is constructed to test research hypothesis H3:

$$HP_{i,t+1} = \beta_0 + \beta_1 ESG_{i,t} + \beta_2 Controls_{i,t} + Industry + Year + \varepsilon_{i,t}$$
$$GIP_{i,t+1} = \beta_0 + \beta_1 ESG_{i,t} + \beta_2 HP_{i,t} + \beta_3 Controls_{i,t} + Industry + Year + \varepsilon_{i,t}$$

The regression results can be found in columns (3) and (4) of Table 7. Column (3) indicates that the coefficient is significantly positive, suggesting that enhancing ESG performance improves human capital levels. In column (4), the coefficient for human capital level is 0.177, which is also significantly positive. This suggests that increasing human capital level promotes the green innovation performance of companies. The coefficient for ESG performance is significantly positive as well, but its magnitude is smaller compared to the coefficient in the baseline regression. This indicates that the direct effect of ESG performance on green innovation performance of companies is smaller than the effect after including human capital, suggesting that the improvement in human capital level plays a

partial mediating role, which aligns with the conclusions of previous studies and validating research hypothesis H3 [55].

### 4.6. Expand Analysis

4.6.1. The Impact of Corporate ESG Performance Sub-Dimensions on Green Innovation Performance

Based on theoretical analysis and research hypothesis H1, all three sub-dimensions of corporate ESG performance have a positive impact on green innovation performance. In this study, we used the sub-scores of environmental performance (E), social responsibility performance (S), and corporate governance performance (G) from the HuaZeng ESG rating as proxy variables. The regression results can be found in Table 8. It can be observed that the coefficients for E, S, and G are all positive and statistically significant, with the magnitude of the coefficients following the order G > E > S. This suggests that all three sub-dimensions of corporate ESG performance positively influence green innovation performance, but to varying degrees. Among them, corporate governance performance has the strongest effect on green innovation.

**Table 8.** The analysis results of the impact of ESG sub-dimensions on green innovation performance.

| | (1) | (2) | (3) |
|---|---|---|---|
| | F.lnGIP | F.lnGIP | F.lnGIP |
| E | 0.00256 * | | |
| | (1.98) | | |
| S | | 0.00162 * | |
| | | (1.99) | |
| G | | | 0.00435 *** |
| | | | (4.09) |
| _cons | −5.343 *** | −5.264 *** | −5.486 *** |
| | (−11.18) | (−11.09) | (−11.49) |
| Controls | Yes | Yes | Yes |
| Industry | Yes | Yes | Yes |
| Year | Yes | Yes | Yes |
| N | 24,262 | 24,262 | 24,262 |
| R2 | 0.206 | 0.206 | 0.207 |

*t statistics in parentheses. * $p < 0.1$, *** $p < 0.01$.*

4.6.2. The Impact of Corporate ESG Performance on Strategic Green Innovation and Substantial Green Innovation

Reference to existing research [56], green innovation can be classified into strategic innovation and substantial innovation based on the motivation for innovation and the "technological content" of innovation. In this study, green invention patents and utility model patents are used as proxy variables [57]. The regression results can be found in Table 9. Compared to the baseline regression results, the absolute values of the ESG coefficients in columns (2) and (3) slightly decrease (reduced to 0.0304 and 0.0245), but remain statistically significant at the 1% level. It can be observed that the impact of ESG performance on green innovation performance primarily comes from green invention patents. Therefore, it can be concluded that ESG advantages facilitate the improvement of both strategic and substantial green innovation performance.

**Table 9.** The analysis results of the impact of ESG performance on different types of green innovation performance.

|  | (1) | (2) | (3) |
|---|---|---|---|
|  | **F.lnGIP** | **F.lnGIia** | **F.lnGUia** |
| ESG | 0.0319 *** | 0.0304 *** | 0.0245 *** |
|  | (4.30) | (4.68) | (3.81) |
| Controls | Yes | Yes | Yes |
| _cons | −5.414 *** | −4.403 *** | −3.820 *** |
|  | (−10.66) | (−9.50) | (−9.72) |
| Industry | Yes | Yes | Yes |
| Year | Yes | Yes | Yes |
| N | 24,262 | 24,262 | 24,262 |
| *R2* | *0.207* | *0.155* | *0.162* |

*t statistics in parentheses. \*\*\* $p < 0.01$.*

4.6.3. The Impact of Corporate ESG Performance on Independent Green Innovation Performance and Collaborative Green Innovation Performance

Considering that corporate ESG performance contributes to collaborative innovation among companies, this study uses the sum of green invention patents and utility model patents (lnGJA) jointly filed by listed companies as a proxy variable for collaborative green innovation performance. The analysis results can be found in Table 10. Column (2) shows that the coefficient of ESG is significantly positive and slightly smaller than the baseline regression results in column (1). It can be observed that corporate ESG performance not only promotes independent innovation among listed companies but also facilitates the pooling of resources and knowledge complementarity, thereby promoting collaborative innovation performance among companies.

**Table 10.** The analysis results of the impact of corporate ESG performance on independent and collaborative green innovation performance.

|  | (1) | (2) |
|---|---|---|
|  | **F.lnGIP** | **F.lnGJA** |
| ESG | 0.0319 *** | 0.0110 * |
|  | (4.30) | (2.25) |
| _cons | −5.205 *** | −1.603 *** |
|  | (−10.99) | (−5.50) |
| Controls | Yes | Yes |
| Industry | Yes | Yes |
| Year | Yes | Yes |
| N | 24,262 | 24,262 |
| R2 | 0.207 | 0.070 |

*t statistics in parentheses. \* $p < 0.1$, \*\*\* $p < 0.01$.*

## 5. Conclusions and Prospects

### 5.1. Research Findings and Discussion

Based on stakeholder theory, signal transmission theory, and principal-agent theory, this study reasoned and discussed the promotion effect of corporate ESG performance on green innovation performance, as well as the mediating role of financing constraints and human capital level. Hypotheses related to these research findings were formulated. Then, these hypotheses were empirically tested, and the results were subjected to endogeneity treatment and robustness tests to enhance the reliability of the conclusions. Finally, the study further explored the green innovation effect of ESG performance in different dimensions of ESG performance and different types of green innovation. The research findings are as follows:

### 5.1.1. Corporate ESG Performance Promotes Green Innovation Performance

According to the benchmark regression results in this study (Section 4.3 Regression Result Analysis), corporate ESG performance positively promotes green innovation performance. Firstly, companies with high ESG performance usually integrate the concepts of green, innovation, and sustainable development into their strategies and culture. The management has a strong environmental awareness and has accumulated a knowledge base of environmental protection and green innovation. This helps to create a green-oriented innovation atmosphere and take practical actions by allocating resources, without the need to start building knowledge from scratch. Secondly, companies with high ESG performance typically pay better attention to and maintain the interests of stakeholders. They establish and maintain stable trust relationships, as well as a responsible corporate image. This not only facilitates the acquisition of heterogeneous and scarce resources for green innovation from key stakeholders but also enables timely and accurate understanding of their needs and preferences through effective communication. Consequently, targeted green innovation can be undertaken. Additionally, it fosters cooperation among companies, expands knowledge sources, and promotes the sharing and complementarity of internal and external knowledge, providing a solid knowledge base for green innovation. Lastly, companies with high ESG performance tend to have well-established corporate governance structures, incentive mechanisms, and supervision arrangements. This benefits in reducing agency problems, avoiding opportunistic and short-term behavior, and also helps to increase the willingness to bear risks and invest in research and development. Green innovation involves significant adjustments in corporate mechanisms and resources, requiring scientific decision-making and management. Good corporate governance ensures the scientific and rational nature of these adjustments, thereby facilitating the smooth implementation and effectiveness of green innovation.

### 5.1.2. Corporate ESG Performance Improves Green Innovation Performance by Mitigating Financing Constraints and Enhancing Human Capital

Based on the results of mechanism test 1 in this study (Section 4.5.1 Mediation Effect of Financial Constraints), corporate ESG performance improves green innovation performance by alleviating financial constraints. Financing constraints are crucial factors that hinder corporate green innovation. Improving ESG performance helps to transmit non-financial information, reduce information asymmetry, enhance corporate transparency and predictability, and build a responsible corporate image. Moreover, good corporate governance helps reduce agency problems. On the one hand, this reduces the risk premium sought by external financiers due to information asymmetry and agency problems, thus lowering financing costs and expanding financing capacity. On the other hand, it enhances investor confidence and secures more stable long-term sources of funding.

According to the results of mechanism test 2 in this study (Section 4.5.2 Mediation Effect of Human Capital Level), corporate ESG performance improves green innovation performance by enhancing human capital levels. Human capital is both a crucial factor for companies to engage in green innovation and an important source of knowledge and capabilities needed for green innovation. Improving ESG performance contributes to the development of human capital. Firstly, corporate ESG performance acts as a positive signal transmitted in the labor market, releasing information about internal working conditions and career development prospects. This reduces information asymmetry and makes companies more competitive in the labor market, attracting high-quality employees with environmental ethics. Secondly, companies with high ESG performance usually safeguard the interests of their employees. This helps to reduce job insecurity, increase job satisfaction, stimulate employees' enthusiasm, and improve their efficiency, consequently promoting employee innovation. Lastly, companies with high ESG performance tend to have improved employee training and career development systems, contributing to the expansion of employees' knowledge, improving their skills, and enhancing their innovation capabilities.

### 5.1.3. Environmental Performance, Social Responsibility Performance, and Corporate Governance Performance All Promote Green Innovation Performance

According to further research 1 in this study (Section 4.6.1 The Impact of Corporate ESG Performance Sub-dimensions on Green Innovation Performance), different sub-dimensions of corporate ESG performance have varying effects on green innovation performance. The three sub-dimensions of corporate ESG performance positively influence green innovation performance, with corporate governance performance having the strongest and most significant effect. A possible explanation for this is that a small number of companies may reduce or slow down green innovation activities after reaching a satisfactory level of environmental performance, as the demand decreases. Similarly, a few companies may be crowded out of green innovation activities due to the allocation of substantial tangible resources to meet stakeholder interests when improving social responsibility performance. However, the resources invested in improving corporate governance, such as optimizing organizational structure and improving systems, are relatively fewer tangible resources and they exert less pressure on the necessary resources for green innovation. The improvement of corporate governance also serves as the basis for sound decision-making and long-term strategic planning, which relates to the level of environmental and social responsibilities, hence optimizing the decision-making and resource allocation for green innovation.

### 5.1.4. Corporate ESG Performance Promotes Both Strategic Green Innovation and Substantive Green Innovation

According to further research 2 in this study (Section 4.6.2 The Impact of Corporate ESG Performance on Strategic Green Innovation and Substantial Green Innovation), corporate ESG performance has different impacts on strategic green innovation and substantial green innovation. In order to meet external pressures, such as complying with government policies, companies sometimes engage only in strategic green innovation to meet quantity and speed requirements, while not strongly inclined to invest significant resources and time in substantive green innovation. However, under the influence of corporate ESG practices, companies not only engage in strategic innovation to achieve short-term benefits but also undertake substantive green innovation to enhance competitiveness and achieve sustainable development. Companies that improve ESG performance integrate the concepts of green development and innovation-driven into their operations and strategies. This enhances managers' awareness of environmental protection and sustainable development, pursuit of long-term benefits and competitiveness, and willingness to engage in substantive green innovation.

### 5.2. Research Implications

#### 5.2.1. Government Governance Recommendations

For China and its government, achieving the dual carbon goals and promoting the high-quality development of listed companies are key tasks in the future. Therefore, the ESG performance of enterprises should be fully utilized to enhance green innovation performance. Firstly, regulatory agencies need to unify the ESG evaluation system as much as possible and provide authoritative guidance on ESG practices for enterprises. Improve the ESG information disclosure system, explore mandatory ESG information disclosure and report auditing, strengthen the supervision of ESG rating agencies, and ensure the comparability, timeliness, and effectiveness of ESG disclosure information through institutional guarantees, thereby driving companies to improve their ESG performance. Secondly, the government should increase support for companies with excellent ESG performance, such as incorporating ESG evaluation into procurement policies, encouraging financial institutions to provide credit support, tax reductions, and fiscal subsidies, etc., to motivate companies to improve their ESG performance. Lastly, local governments should improve talent policies, solve the problem of attracting talent and improve the quality of public services to attract high-quality talents and create a talent pool, allowing ESG performance

to effectively attract talents and provide sufficient sources of green innovation talents for enterprises.

### 5.2.2. Investor Decision-Making Recommendations

For investors, their ultimate goal is to increase investment returns. Under the guidance of ESG concepts, investors are also beginning to pay attention to responsible investment. The ESG performance of companies represents not only their level of environmental and social responsibility and sustainable development but also indicates their future human capital level and green innovation performance, which in turn forms the company's sustainable competitiveness and brings long-term economic benefits. Therefore, investors should include non-financial information such as ESG performance in their decision-making scope, improve the scientific and rationality of investment decision-making, thereby increasing long-term investment returns, and promoting market selection to improve the overall quality of listed companies. Additionally, considering that ESG disclosure construction is still in progress, behaviors like "greenwashing", voluntary disclosure, and the uncertainty of ESG ratings may lead to deviations in measuring ESG performance. Therefore, investors should pay attention to the sources of ESG performance information and strengthen their judgment and usage.

### 5.2.3. Corporate Management Recommendations

For enterprises, they should first develop an ESG strategy that integrates ESG concepts into their strategies, culture, daily operations, and implement them into various levels of management systems. This ensures the persistence of ESG work and promotes the systematic development of green innovation. Secondly, enterprises should improve corporate governance to enhance the scientific and rational decision-making of ESG and green innovation. Investment in ESG and green innovation should be increased, and various resources such as manpower and financial resources should be allocated for related work. This will help facilitate the in-depth development of ESG and innovation work. Lastly, enterprises should improve the quality of ESG information disclosure, introduce third-party verification to enhance credibility, and engage in effective communication of ESG information through diverse means to improve its understandability.

**Author Contributions:** Conceptualization, J.Z.; methodology, J.Z.; software, J.Z.; validation, J.Z.; formal analysis, J.Z.; investigation, J.Z.; resources, J.Z.; data curation, J.Z.; writing—original draft preparation, J.Z.; writing—review and editing, J.Z.; visualization, J.Z.; supervision, Z.L.; project administration, J.Z. All authors have read and agreed to the published version of the manuscript.

**Funding:** This research received no external funding.

**Institutional Review Board Statement:** Not applicable.

**Informed Consent Statement:** Informed consent was obtained from all subjects involved in the study.

**Data Availability Statement:** The data presented in this study are available on public dataset.

**Conflicts of Interest:** The authors declare no conflict of interest.

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
