# Peer review of "Study on the Impact of Corporate ESG Performance on Green Innovation Performance—Evidence from Listed Companies in China A-Shares"

_sustainability, doi:10.3390/su152014750_

Round 1

Reviewer 1 Report

Authors focused on the analyzes of the green innovation effect of corporate ESG performance using Chinese A-share listed companies as a sample from 2009 to 2021. The research shows that corporate ESG performance can enhance green innovation performance. Mechanism analysis reveals that ESG performance mainly improves green innovation performance by alleviating financing constraints and enhancing human capital. Further research shows that all three sub-dimensions of ESG performance contribute to improving green innovation performance, with the strongest effect observed in corporate governance performance. The subject matter is extremely important nowadays, as it not only enhances strategic green innovation performance and independent green innovation performance but also improves substantial green innovation performance and collaborative green innovation performance.

Comments and Suggestions for Authors

·   Line 14: ESG – the full name should be entered in parentheses (Environmental, Social, Governance) when the parameter is mentioned for the first time.

·         Line 43: Lack of space between „neutrality” and parenthesis.

·         Line 55: Lack of space between „innovation” and parenthesis.

·         Line 69: Lack of space between „improved” and parenthesis.

·         Line 105: Lack of space between „innovation” and parenthesis.

·         Line 108: Lack of space between „innovation” and parenthesis.

·         Line 117: Lack of space between „stakeholders” and parenthesis.

·         Line 127: Lack of space between „performance” and parenthesis.

·         Line 137: Lack of space between „diversity” and parenthesis.

·         Line 138: Lack of space between „plans” and parenthesis.

·         Line 156: Lack of space between „straints” and parenthesis.

·         Line 206: CICC – the full name should be entered in parentheses.

·         Line 209: CSMAR – the full name should be entered in parentheses.

·         Line 221: MSCI – the full name should be entered in parentheses.

·         Line 253: Lack of space between „variables” and parenthesis.

·         Line 277: “correlation” should be “Correlation”.

·         Table 3. Reducing the distances between variables will also reduce the size of the table and make it more readable.

·         Line 307: Lack of space between „performance” and parenthesis.

·         Line 347: Lack of space between „activities” and parenthesis.

·         Line 420: Lack of space between „research” and parenthesis.

·         Line 423: Lack of space between „variables” and parenthesis.

·         Line 431: Description of Table 9 should be “The analysis results of the impact of ESG performance on different types of green innovation performance”.

·         Line 446: Description of Table 10 should be should be “The analysis results of the impact of corporate ESG performance on independent and collaborative green innovation performance”.

Author Response

Thank you very much for the professional and meticulous editing suggestions. Based on the suggestions provided by the editor, we have made revisions to each item, as follows:

 Line 14: ESG – the full name should be entered in parentheses (Environmental, Social, Governance) when the parameter is mentioned for the first time.

-Modified

  • Line 43: Lack of space between „neutrality” and parenthesis.

-Modified

  • Line 55: Lack of space between „innovation” and parenthesis.

-Modified

  • Line 69: Lack of space between „improved” and parenthesis.

-Modified

  • Line 105: Lack of space between „innovation” and parenthesis.

-Modified

  • Line 108: Lack of space between „innovation” and parenthesis.

-Modified

  • Line 117: Lack of space between „stakeholders” and parenthesis.

-Modified

  • Line 127: Lack of space between „performance” and parenthesis.

-Modified

  • Line 137: Lack of space between „diversity” and parenthesis.

-Modified

  • Line 138: Lack of space between „plans” and parenthesis.

-Modified

  • Line 156: Lack of space between „straints” and parenthesis.

-Modified

  • Line 206: CICC – the full name should be entered in parentheses.

-Modified

  • Line 209: CSMAR – the full name should be entered in parentheses.

-Modified

  • Line 221: MSCI – the full name should be entered in parentheses.

-Modified

  • Line 253: Lack of space between „variables” and parenthesis.

-Modified

  • Line 277: “correlation” should be “Correlation”.

-Modified

  • Table 3. Reducing the distances between variables will also reduce the size of the table and make it more readable.

-Modified

  • Line 307: Lack of space between „performance” and parenthesis.

-Modified

  • Line 347: Lack of space between „activities” and parenthesis.

-Modified

  • Line 420: Lack of space between „research” and parenthesis.

-Modified

  • Line 423: Lack of space between „variables” and parenthesis.

-Modified

  • Line 431: Description of Table 9 should be “The analysis results of the impact of ESG performance on different types of green innovation performance”.

-Modified

  • Line 446: Description of Table 10 should be should be “The analysis results of the impact of corporate ESG performance on independent and collaborative green innovation performance”.

-Modified

To facilitate the review process for the reviewer and editor, we have made revisions in track changes mode. Once again, we sincerely appreciate the reviewer's meticulous and professional feedback. Wishing you a pleasant day!

Reviewer 2 Report

Review for the article titled Study on the Impact of Corporate ESG Performance on Green Innovation Performance - Evidence from Listed Companies in  China A-shares

The authors write on an extremely timely topic which explores the impact of corporate ESG (Environmental, Social, and Governance) performance on green innovation in Chinese A-share listed companies from 2009 to 2021. The research finds that ESG performance positively influences green innovation, primarily by reducing financing constraints and enhancing human capital. It also shows that all three aspects of ESG performance (environmental, social, and governance) contribute to improving green innovation, with corporate governance having the most significant effect.

General Comments:

The paper is well written with minimal-moderate language  errors which can be edited on the revised version

Model Specification

Pease address my concerns below:

1.      The model assumes a linear relationship between ESG factors and green innovation performance. In reality, this relationship may not be linear, and there could be complex interactions and thresholds. How do the authors justify a liner model?

2.      Green innovation performance is measured solely by the sum of green invention patents and utility model patents. This narrow focus on patent counts may not capture the full spectrum of green innovation activities, such as process improvements, product design changes, or sustainability initiatives that don't result in patents. Please justify this narrow limitation further.

3.      ESG factors are multidimensional and complex. The econometric specification aggregates ESG factors into a single variable (ESG). However ESG factors are complex and multidimensional.. Different ESG subcomponents may have varying effects on green innovation. Please justify.

References

Please include fmore recent articles in your study. ESG has gained attention in literature a lot of publications have been produced. Please credit more leading studies in the domain. An additiona 7-10 studies will suffice to be enough.

mim - moderate editing

Author Response

We greatly appreciate the professional and meticulous comments from the reviewers. We have carefully considered the reviewers' comments and have made the following modifications and responses accordingly.

Regarding question 1:Regarding the main causal relationship discussed in this paper, whether it is linear or not, we provide the following responses to the reviewer:

1) Before conducting the empirical analysis in this study, we performed a graph fitting analysis on the data. Specifically, we plotted scatter plots for the core explanatory variable and the dependent variable, and based on the visual analysis, we preliminarily determined that they exhibit a linear relationship. Subsequently, we used the SPSS software to fit the scatter plots, considering linear, quadratic, logarithmic, and exponential models. The results showed that the linear fitting had the best performance. In order to make the paper more concise and follow the general paradigm of empirical economic research papers, we did not discuss this step in the paper.

2) After constructing the model, we estimated the model based on the data. The regression results supported our research hypothesis, and the main effect was statistically significant at the 1% level, indicating the justification of constructing a linear model.

3) Furthermore, to ensure the robustness of the results, we further estimated the model using instrumental variable methods, substituted explanatory variables, alternative estimation models, substituted dependent variables, and controlled for exogenous shocks. The results passed the significance tests, further demonstrating the validity of the model construction.

Regarding question 2: Regarding the use of patent quantity as the measure of green innovation performance in this paper, we acknowledge that it may not represent the entire scope of green innovation activities, as mentioned by the reviewer. Indeed, green innovation performance includes various factors, such as process improvements and product design changes, as pointed out by the reviewer, and using a single variable may overlook certain aspects of green innovation performance. However, there are two reasons why we chose to use patent quantity: 1) In the data collection process, we need to consider the availability of data. Among the data that can be collected, patent quantity is the indicator that is most closely related to its meaning. Therefore, in this paper, patent quantity is used as a proxy for green innovation performance. 2) At the same time, we have also referred to multiple related articles on the topic and found that many papers adopt patent quantity as a measure of green innovation performance. This provides reference for measuring green innovation performance in this study.

Furthermore, we understand the reviewer's concerns. Therefore, in the revised manuscript, we have added references to other literature that uses patent quantity as a measure of green innovation performance in the section '3.2.1 Dependent Variable' to enhance the persuasiveness.

Regarding question 3: Regarding the issue of how different sub-components of ESG may have varying impacts on green innovation, indeed, as the reviewing teacher pointed out, ESG encompasses various factors, and aggregating multiple factors together may overlook the influence of individual sub-components on green innovation. Therefore, in the section "4.6.1 The Impact of Corporate ESG Performance Sub-dimensions on Green Innovation Performance" of this paper, ESG is further divided into environmental performance (E), social responsibility performance (S), and corporate governance performance (G), and the impact of corporate ESG on different sub-dimensions is explored. Additionally, in sections 4.6.2 and 4.6.3, the paper further subdivides innovation performance into strategic innovation and substantive green innovation, independent green innovation performance and collaborative green innovation performance, in order to enrich the conclusions of this paper.

To facilitate the review process for the reviewer and editor, we have made revisions in track changes mode. Once again, we sincerely appreciate the reviewer's meticulous and professional feedback. Wishing you a pleasant day!

Reviewer 3 Report

Dear Authors,

First and foremost, thank you very much for your submission, I really enjoyed reading your paper, i believe that the topic is of utmost import, however I have following points that I would like to see addressed:

1- I understand that your sample is winsorized, however, if the sample was not winsorized what would be the impact on the analysis?

2- Do you think the result could not be better using another type of regression strategy?

3- I'd like you to address the possibility to extend the approach to other markets? Do you think we could expect similar results or are those marketwise.

4- Some elements of the conclusion are loosely related to the quantitative analysis, would you mind clearly state what elements of the quantitative analysis allow you to write each and every statement?

I believe that your paper is interesting and has some real quality, however I'd like those point to be addressed before considering it for publication.

Best Regards,

Your reviewer

Moderate editing

Author Response

Thank you very much for the professional and meticulous editing suggestions. Based on the suggestions provided by the editor, we have made revisions to each item, as follows:

Regarding question 1:We understand the reviewer's concern. During the writing process of this paper, we did not perform truncation treatment during the initial regression estimation. The significance level and the results after truncation treatment were consistent, both significant at the 1% level, with only slight differences in the regression coefficients. This does not affect our hypothesis testing. However, we decided to perform truncation treatment for the following two reasons:

1) Before the preliminary regression estimation, we plotted a scatter plot of the sample data and observed a few extreme outliers. In the data cleaning process, we needed to remove such samples to ensure the accuracy of our regression estimation.

2) As mentioned in the paper, the data for this study spanned 12 years and included information from 3,877 companies. The sample size was sufficiently large, allowing us to perform truncation treatment without concern of distorting the results. Additionally, we cautiously chose a 1% truncation level to minimize sample loss.

Regarding question 2:This is a very worthy question to discuss. In this paper, we chose to use a panel fixed effects model for regression estimation based on the following reasons:

    1) Suitability of data characteristics: The corporate data used in this paper is panel data, involving observations from multiple companies over multiple time periods. Using a panel fixed effects model helps better control for individual fixed effects, eliminate heterogeneity between individuals, and improve the accuracy of the model, thus reducing endogeneity issues caused by individual heterogeneity.

2) Relevance to the research question: The main focus of this paper is to explore the impact of corporate ESG performance on green innovation performance. The fixed effects model is suitable for controlling for individual fixed effects and investigating the effects of explanatory variables on the dependent variable, which aligns well with the research question in this paper.

3) Reference to relevant literature: In selecting the model, we referred to several similar studies and found that the fixed effects model was well-suited for the research question in this paper. These three reasons justify our use of the fixed effects model. However, we acknowledge that there are other models mentioned in similar studies, such as the 'Bilateral Stochastic Frontier Model' used by Zhu et al, 2023. Based on our data characteristics and the main research question, the panel fixed effects model is likely more appropriate for this paper. Nevertheless, we understand the reviewer's concern, and to address it, we have added references to the literature on model selection in the revised manuscript, enhancing its persuasiveness.

Regarding question 3: About the possibility to extend the approach to other markets.The fixed effects model is widely applied in economic research for panel data analysis and exhibits strong generalizability. The model is not only applicable to the current research market but also to similar markets. This is because the fixed effects model, by controlling for individual fixed effects, eliminates individual heterogeneity, thereby improving the accuracy and robustness of the model. Therefore, it is reasonable and feasible to extend the fixed effects model to other market theories. However, it is important to note the particularities and differences of different markets, as structural and characteristic differences among markets may have an impact on research findings. Thus, when extending to other markets, appropriate adjustments and modifications may be needed to adapt to the new market conditions. Gathering data from different markets, comparing the model estimation results across different markets, is crucial to validate the applicability of generalizability.

Regarding question 4:About Some elements of the conclusion are loosely related to the quantitative analysis.Indeed, as the reviewer mentioned, the connection between the conclusion section and the quantitative analysis in the previous sections of this paper is not sufficiently tight. In response to this feedback, we have made improvements in the revised manuscript by ensuring a more coherent link between the conclusion points and the empirical findings. In the revised version, we have made sure that each conclusion point corresponds to the relevant empirical results.

To facilitate the review process for the reviewer and editor, we have made revisions in track changes mode. Once again, we sincerely appreciate the reviewer's meticulous and professional feedback. Wishing you a pleasant day!

References:

Sun H, Zhang B, Yang Z, Xia X. Bilateral Effects of ESG Responsibility Fulfillment of Industrial Companies on Green Innovation. Sustainability. 2023; 15(13):9916. https://doi.org/10.3390/su15139916

Reviewer 4 Report

Τhis is a very interesting manuscript.

The authors have provided a detailed analysis of the research assumptions, the data used, the model description, the results obtained and their implications. 

The paper's quality would be further upgraded if:

a) a literature review was added, surveying the relevant studies conducted previously for China and other countries, the methodology and variables used as proxies and the results obtained

b) the results of the study were enhanced by comparing them to those of the relevant studies presented in the literature review

Author Response

Thank you very much for the professional and meticulous editing suggestions. Based on the suggestions provided by the editor, we have made revisions to each item, as follows:

1) Regarding the issue of expanding the literature review, based on the advice of our reference teacher, we have expanded our literature review in Section 2.1 and added literature citations in the model selection and other sections. After the modifications, we have included an additional 11 references, aiming to enrich the theoretical part of the article and enhance its persuasiveness.

2) Regarding the comparison of research results with existing literature, we have added comparisons with existing studies in different sections such as Section 4.3, which focuses on benchmark regression, and Section 4.5, which involves mechanism testing. Through these comparisons, we aim to highlight the research findings of our article.

To facilitate the review process for the reviewer and editor, we have made revisions in track changes mode. Once again, we sincerely appreciate the reviewer's meticulous and professional feedback. Wishing you a pleasant day!

Round 2

Reviewer 3 Report

Dear Authors,

thank you very much for addressing the questions raised, considering the potential loss of information related to winsorizing, i would recommend at least putting a sentence acknowledging the issue in particular considering that their are noises and information asymetry problems (you could add in the bibliography the work of

Hassani BK, Bahini Y. Relationships between ESG Disclosure and Economic Growth: A Critical Review. Journal of Risk and Financial Management. 2022; 15(11):538. https://doi.org/10.3390/jrfm15110538

whom address the matter.

Once this point has been addressed, this paper will be suitable for publication.

Best Regards,

Bertrand

None

Author Response

Dear reviewer,

Thank you once again for your meticulous and professional suggestions. As you mentioned, scholars do have different opinions on this issue in similar literature. We should avoid attributing these differences to data truncation. Therefore, in the second round of revisions, we have provided more detailed explanations on the truncation process and included the reference that you suggested to enhance the persuasiveness of our argument.

Thank you again for your suggestions, and I hope you have a pleasant life.

Round 3

Reviewer 3 Report

All elements have been addressed i am happy to recommend that paper for publication.

Just minor elements 

Author Response

We greatly appreciate your suggestions for the revision of our paper. Your input has made our paper more comprehensive. Thank you and have a pleasant life.